# Anthropometric Profile and Physical Activity Level as Predictors of Postural Balance in Overweight and Obese Children

**DOI:** 10.3390/bs13010073

**Published:** 2023-01-14

**Authors:** Eduardo Guzmán-Muñoz, Guillermo Mendez-Rebolledo, Cristián Núñez-Espinosa, Pablo Valdés-Badilla, Matías Monsalves-Álvarez, Pedro Delgado-Floody, Tomás Herrera-Valenzuela

**Affiliations:** 1Escuela de Kinesiología, Facultad de Salud, Universidad Santo Tomás, Talca 3480094, Chile; 2Escuela de Kinesiología, Facultad de Ciencias de la Salud, Universidad Autónoma de Chile, Talca 3480094, Chile; 3Escuela de Medicina, Universidad de Magallanes, Punta Arenas 6200000, Chile; 4Centro Asistencial Docente y de Investigación, Universidad de Magallanes, Punta Arenas 6200000, Chile; 5Departamento de Ciencias de la Actividad Física, Facultad de Ciencias de la Educación, Universidad Católica del Maule, Talca 3480094, Chile; 6Carrera de Entrenador Deportivo, Escuela de Educación. Universidad Viña del Mar, Viña del Mar 2520000, Chile; 7Instituto de Ciencias de la Salud, Universidad de O’Higgins, Rancagua 2820000, Chile; 8Department of Physical Education, Sport and Recreation, Universidad de La Frontera, Temuco 4780000, Chile; 9Department Physical Education and Sports, Faculty of Sport Sciences, University of Granada, 18001 Granada, Spain; 10Escuela de Ciencias de la Actividad Física, el Deporte y la Salud, Facultad de Ciencias Médicas, Universidad de Santiago de Chile (USACH), Santiago 8320000, Chile

**Keywords:** postural control, adiposity, pediatric obesity

## Abstract

Overweightness and obesity can negatively influence many activities, including postural balance and locomotion, increasing predisposition to injury and risk of falls due to limitations on the biomechanics of daily living. The present study aimed to determine the influence of the anthropometric profile and physical activity level (PAL) on the postural balance of overweight and obese children. The sample included 387 schoolchildren (216 boys and 171 girls). The variables of the anthropometric profile studied were body mass, biped height, BMI, waist circumference, waist-to-hip ratio (WHR), summation of folds, body composition, and somatotype. PAL was measured using the PAQ-C questionnaire. Static and dynamic postural balance were measured through an open-eye (OE) and closed-eye (CE) posturographic test and the SEBTm, respectively. For static balance, the significant models were for mediolateral velocity (R^2^ = 0.42 in OA; R^2^ = 0.24 in OC), anteroposterior velocity (R^2^ = 0.21 in OA; R^2^ = 0.27 in OC), and mean velocity (R^2^ = 0.27 in OA; R^2^ = 0.46 in OC), where the predictors of low performance were younger age, male sex, overweight/obese nutritional status, greater thickness of skin folds, less tendency to mesomorphy, and greater fat mass. On the other hand, for dynamic postural balance, the significant models were observed in the previous direction (R^2^ = 0.39), posteromedial (R^2^ = 0.57), and posterolateral (R^2^ = 0.56), where the variables that predict a low performance were low PAL, overweight/obese nutritional status, and high WHR. Overweight and obese children presented a deficit in static and dynamic postural balance, enhanced by variables such as gender, age, PAL, and anthropometric characteristics related to adiposity.

## 1. Introduction

Overweightness and obesity are defined as abnormal and excessive accumulations of fat that can be detrimental to health and manifest in excess weight and body volume [1]. Childhood overweightness and obesity have increased over the last few decades: globally, the World Obesity Atlas 2022 recently estimated that 13% of children are overweight and 11% are obese [1]. In Europe, overweightness and obesity in schoolchildren are close to 20% and increasing in males [2]. In Latin America, the situation remains substandard or worse, showing close to 50% of overweightness and obesity in children between 5–9 years of age [3]. This increase in childhood obesity is worrying since, due to projections, approximately 55% of obese children go on to be obese in adolescence; around 80% of obese adolescents will still be obese in adulthood, and around 70% will be obese over age 30 [4].

Obesity can negatively influence many activities, including postural balance and locomotion, increasing predisposition to injury and the risk of falls due to limitations on the biomechanics of daily living [5]. Obesity also seems to be a factor that contributes to reducing the efficiency of obese individuals when performing motor tasks in bipedal postures, presumably due to postural balance restrictions caused by excess weight [6]. Overweight and obese children walk with a wider base of support and increase the time they spend in contact with the ground, decreasing the step length and increasing the duration of the double support phase to maintain postural balance while walking [7].

As an inherent factor of movement, proper postural balance is crucial for motor development in general and the good performance of the functional activities of daily living [8]. In overweight and obese children, it has been reported that postural balance is decreased compared with normal-weight children [9]. However, few reports associate anthropometric measurements and body composition with postural balance deficits observed in children with excess weight. In this context, it can be seen that variables such as height [10], arm length [10], foot length [10], body mass index [11], and body fat percentage [11] have been correlated to postural balance. For these reasons, the possible influence of anthropometric and body composition factors on postural balance performance is relevant, and, consequently, so is the role of postural balance in the acquisition and development of fundamental motor skills.

In children and adolescents, one of the main risk factors for overweightness and obesity is reduced physical activity and a sedentary lifestyle [12]. The genetic predisposition for a high BMI is between 25% and 40%, suggesting the vast potential of environmental influences, including exposure to physical activity, on the development or prevention of obesity [12]. Studies have shown an inverse relationship between the physical activity level (PAL) and BMI in children [12,13], emphasizing the critical role of physical activity in childhood obesity. In addition, it can be seen that a low PAL could negatively influence the performance of motor tasks [14]. However, results have also been reported indicating that physical activity and motor skills develop independently in early childhood [15].

Due to the importance of postural balance in the acquisition and development of fundamental motor skills for human movements, such as standing, walking, and jumping, this research aimed to determine the influence of the anthropometric profile and PAL on the postural balance of overweight and obese children.

## 2. Materials and Methods

### 2.1. Design

This study featured a cross-sectional design and was reported following the STROBE statement. The research complied with all the relevant national regulations and institutional policies. All participants and their parents signed a consent term following the Declaration of Helsinki, which was approved by the local Ethics Committee (registration number 212017).

### 2.2. Participants

The sample for this research corresponds to 387 Caucasian schoolchildren (216 boys and 171 girls) between 6 and 9 years old belonging to different public educational establishments in the commune of Talca, Maule Region (Chile). Participants from 8 public schools were selected under a probabilistic stratified proportional sampling by sex and primary education courses (first, second, third, and fourth grade). The ideal sample calculated was 361 schoolchildren between 6 and 9 years old, considering a population of 5958 schoolchildren according to data obtained from the Maule region ministerial Secretary of Education. A margin of error of 5% and a confidence level of 95% were used for this calculation. Participants with the following conditions were excluded: (a) those who presented some neurological and/or vestibular alterations; (b) any musculoskeletal injuries, such as fractures, sprains, dislocations, or muscle tears during the six months before evaluation; (c) the presence of any inflammatory or painful conditions at the time of evaluation; (d) uncorrected visual impairment; and (e) the use of technical assistance for walking.

### 2.3. Anthropometric Profile

To carry out the evaluations, the recommendations of the International Society for Advances in Kinanthropometry (ISAK) were followed. Bipedal height was first assessed with a stadiometer (Seca, Hamburg, Germany; precision 0.1 cm) and body weight with a digital scale (Seca, Hamburg, Germany; precision 0.1 kg). BMI was calculated by dividing body weight (kg) by standing height squared (m^2^). Subsequently, the diameters were evaluated with anthropometers (Rosscraft, Canada, precision 0.1 mm), the perimeters with a tape measure (Sanny, Brazil, precision 0.1 mm), and the skinfolds with a caliper (Harpenden, England, precision 0, 2 mm). Anthropometric measurements consisted of six diameters (biacromial, transverse thorax, anteroposterior thorax, bicrestal, biepicondylar humerus, and biepicondylar femur), ten girths (head, relaxed arm, flexed arm in tension, maximum forearm, mesosternal thorax, minimum waist, maximum hip, maximal thigh, medial thigh, and maximal calf), and six skinfolds (triceps, subscapular, supraspinal, abdominal, medial thigh, and maximal calf). These evaluations made it possible to obtain the body composition as proposed by Kerr (Kerr, 1988), which establishes five components (pentacompartmental method): adipose, muscle, residual, bone, and skin mass. On the other hand, the somatotype was determined according to Heath and Carter (Heath and Carter, 1967), who defined the quantification of the shape and composition of the human body through three numbers represented by endomorphy, mesomorphy, and ectomorphy.

### 2.4. Physical Activity Level (PAL)

The PAL was measured through the PAQ-C. The PAQ-C is a questionnaire that measures general moderate–vigorous physical activity levels over the last seven days during the school year. It consists of ten questions with response options on a five-point scale. The questions inquire about activities carried out during free time and physical activities carried out in physical education classes, during recess, lunch, after school, at night, and on weekends [16]. Question ten is not used as part of the total score but instead to identify students who had unusual activity during the previous week [16]. To calculate the final score, the mean of the 9 questions is estimated, where 1 indicates low physical activity and 5 indicates high physical activity [17]. This questionnaire has been previously validated in Spanish and used on Chilean children [18]. The internal consistency reliability coefficient obtained in our data for this instrument was 0.77.

### 2.5. Static Postural Balance

Static postural balance measurements were carried out on a force platform (Artoficio Ltd.a., Santiago, Chile). All individuals participated in the measurement of static postural balance under two different consecutive conditions: the double-leg stance with eyes open (EO) and eyes closed (EC). Participants were instructed to remain as still as possible for 30 s with arms at the side of the body, feet in line with their shoulders, and looking at a marker located on a wall three meters away and at eye level [19]. All participants were evaluated barefoot and were previously instructed to maintain a symmetrical load with their lower limbs on the platform. In each condition, three trials were performed, and the average of these was used to obtain the COP variables. Four COP variables were processed using MATLAB r2012a (MathWorks Inc., Natick, MA, USA): area, mean velocity, AP velocity, and ML velocity. The intraclass correlation coefficient obtained for this instrument was 0.98 intraday and 0.97 interday.

### 2.6. Dynamic Postural Balance

Dynamic balance was assessed with the modified star excursion balance test (mSEBT). The anterior, posteromedial (PM), and posterolateral (PL) reach directions were evaluated in the dominant leg. The test maintained a single-leg stance while the contralateral limb reached the maximum distance along a line marked on the ground. The participants had to keep their hands on their waists during this task. The line should be touched only with the toes [20]. All reach trials began with both feet in contact with the ground and the stance leg appropriately positioned at about the center of the mSEBT grid. The evaluation was considered valid when the child did not lift the support foot off the ground and managed to return the limb to the initial position without losing balance. A total of three test reaches were performed in each direction. Reach distance was quantified by measuring the distance (in centimeters) from the center of the crosshairs to the point of distal foot–ground contact. Performance in the mSEBT for each direction was represented by the average of the three best reach distances and normalized to limb length. The intraclass correlation coefficient obtained for this instrument was 0.91 intraday and 0.89 interday.

### 2.7. Statistical Analysis

Data were analyzed with SPSS 23.0 statistical software (SPSS 23.0 for Windows, SPSS Inc., Chicago, IL, USA). The mean and standard deviation were calculated to describe the characteristics of the sample: the anthropometric profile, PAL, and postural balance. To determine the reliability of the physical activity self-report questionnaire, Cronbach’s alpha coefficient was used. In contrast, the intraclass correlation coefficient was used to determine the intra- and interday reliability of the postural balance instruments. A multiple linear regression model (95% confidence interval) was applied to determine the influence of the anthropometric profile, nutritional status, and PAL on the variables of static and dynamic postural balance. For this analysis, nutritional status (normal weight/excess weight) and sex (boys/girls) variables were dichotomized. The goodness of fit was determined using the R^2^ coefficient. A collinearity diagnosis was made for each of the variables present in the regression models obtained, where variables with values less than 0.10 tolerance and values above 10.0 variance inflation factor (VIF) were eliminated. The level of significance for all statistical tests was <0.05.

## 3. Results

Table 1 shows the characteristics of the sample in which the mean and standard deviations of the variables age, PAL, anthropometric measurements, body composition, and the somatotype of the students evaluated are described. The BMI of the normal-weight schoolchildren was 15.86 ± 0.74, 18.47 ± 0.87 for the overweight schoolchildren, and 22.04 ± 1.74 for the obese schoolchildren. The results of the static and dynamic balance tests are shown in Table 2.

### 3.1. Linear Regression Analysis for Static Postural Balance Variables in Eyes Open Test

The variables that were significant in the models are shown in Table 3. For ML velocity, the model (R^2^ = 0.422) indicates that younger children (β = −0.231; *p* = 0.001), males (β = 0.227; *p* = 0.037), children with overweight/obese weight status (β = 0.192; *p* = 0.047), greater skinfold thickness (β = 0.014; *p* = 0.013), and children with a higher percentage of adipose mass (β = 0.063; *p* = 0.010) present a lower postural balance. The AP velocity yielded a model (R^2^ = 0.211) that indicates that younger children (β = −0.074; *p* = 0.039) and males (β = 0.133; *p* = 0.024) present poor postural balance. For mean velocity, the model (R^2^ = 0.274) indicates that younger children (β = −0.005; *p* = 0.022), males (β = 0.013; *p* = 0.006), and overweight/obese children (β = 0.014; *p* = 0.003) showed less postural control. The COP area variable did not present significant models.

### 3.2. Linear Regression Analysis for Static Postural Balance Variables in Eyes Closed

The variables that were significant in the models are shown in Table 4. For ML velocity, the model (R^2^ = 0.243) shows that younger children (β = −0.075; *p* = 0.238) with overweight/obese nutritional status (β = 0.238; *p* = 0.003) have less postural balance. The AP velocity showed a significant model (R^2^ = 0.269) that indicates that children with overweight/obesity status (β = 1.000; *p* = 0.003) and less tendency to the ectomorph profile (β = −0.605; *p* = 0.010) present higher values of COP, that is, poor postural balance. For mean velocity, the model (R^2^ = 0.461) indicates that children with overweight/obesity status (β = 0.049; *p* = 0.001), greater adipose mass (β = 0.008; *p* = 0.013), greater tendency to an endomorphic profile (β = 0.010; *p* = 0.033), and less tendency to an ectomorphic profile (β = −0.012; *p* = 0.040) are those with less postural balance. The COP area variable did not present significant models.

### 3.3. Linear Regression Analysis for Dynamic Postural Balance Variables

The variables that were significant in the models are shown in Table 5. For the anterior direction, the model (R^2^ = 0.389) shows that children with low PAL (β = 5.161; *p* = 0.009) and an overweight/obese nutritional status (β = −2.713; *p* = 0.033) have less postural balance. The PM direction of the mSEBT yielded a model (R^2^ = 0.565) that indicates that children with low PAL (β = 6.544; *p* = 0.013) and higher WHR (β = −13.947; *p* = 0.010) have less PM reach, that is, poor postural balance. For the PL direction, the model (R^2^ = 0.564) shows that children with a low PAL (β = 12.540; *p* = 0.001) and overweight/obese status (β = −6.545; *p* = 0.031) are the ones with the least postural balance.

## 4. Discussion

From models obtained through a multiple linear regression analysis, the results of this study reveal that some variables of the anthropometric profile, nutritional status, and PAL predict the performance of static postural balance (variables of the COP) and dynamic variables of the mSEBT in children between 6 and 9 years old. Specifically, for static postural balance, the predictors of poor performance were younger age, male gender, overweight/obese nutritional status, greater skinfold thickness, greater fat mass, less tendency to ectomorphy, and greater tendency to an endomorph somatotype. The variables that predict a poor postural balance for dynamic postural balance were a low PAL, an overweight/obese nutritional status, and a high WHR. Similar findings have previously been reported, demonstrating that nutritional status is correlated to static balance performance [21,22,23]. However, the interaction of other factors, such as body composition and somatotype, which could affect motor performance, was not considered. Dynamically, few studies have analyzed postural balance in overweight and obese children, reporting that excess weight could be correlated with poor balance performance [24,25]. In this research, it was possible to determine that the factors analyzed individually in previous reports, when considered together, can predict postural balance.

This research revealed that measures of adiposity accumulation (higher BMI, thickness of skinfolds, higher percentage of adipose mass, and high WHR) could be, collectively, a determining factor in predicting and explaining poor performance in postural balance. Previous studies have reported correlations between increased BMI and impaired static postural balance in children [11,26]. The COP variables AP velocity, velocity mean, and ML velocity in EO and EC have also been related to BMI [11,26]. However, the interaction between anthropometric variables with age, gender, and nutritional status had not been studied previously through multiple linear regression models. It has also been proposed that the accumulation of fatty tissue around and within the muscle could alter the normal mechanisms of motor responses due to physiological and neuromuscular changes [27]. Some authors have hypothesized that the altered myoelectric manifestations related to poor motor behavior are a response of the central nervous system to electrochemical imbalances in muscle fiber and due to reductions in the propagation velocity of intracellular action potential [28]. A significant relationship has been observed between fat accumulation and the expression of proinflammatory cytokines in muscle, which could reduce electrochemical balance and neural conductivity [29]. Likewise, it can be seen that overweight and obese individuals present alterations in both anticipatory and compensatory muscle activation patterns [30]. This would directly affect the muscular response due to lower neuromuscular efficiency in the recruitment of motor units. It is likely that the overweight and obese children evaluated in this study and their anthropometric variables related to adiposity accumulation negatively influenced the performance of the balance tests.

Another exciting finding in our results was the influence of the PAL on the models obtained in each of the directions of the mSEBT. Although the overweight/obese nutritional status is the associated factor present in each model obtained from the static and dynamic balance, a low PAL could enhance the deterioration of balance in dynamic activities. It is generally assumed that physical activity is causally related to functionality, suggesting that more active individuals usually have better physical fitness and motor abilities [14]. However, little research describes the functional consequences of a low PAL in children. Children with low PALs have been reported to have poor development of motor skills such as object control, precision, coordination, and postural balance [31,32]. Furthermore, it has been found that children with low PALs have less capacity for the development of motor skills such as walking, running, and jumping [33,34,35]. This would directly influence performance in dynamic evaluation tests such as the mSEBT. Our linear regression model showed that the PAL was more influential than nutritional status in the performance of dynamic postural balance, which is related to what was previously reported, where it was indicated that a low PAL can be riskier than excess weight in the physical function and health of people [36].

In this research, gender was a determining factor in the models obtained for static postural balance, with males presenting a worse performance. This, together with an overweight/obese nutritional status, enhances poor balance test performances. Similar results were previously reported, where it was determined that nutritional status and gender influence the deterioration of postural balance in children, with overweight and obese males being the most affected [23]. In the same way, age also became a factor that influences static postural balance, with younger children presenting poor performance. It has been shown that women have a better balance at ages younger than 11–12 years than men [37]. It is believed that the greater motor hyperactivity observed in men in this age group could be the cause of a delay in the maturation of the vestibular system, which could directly affect the development of postural balance [38]. This would explain why gender and age determine static postural balance factors in overweight and obese children.

Just as genetic factors can predispose one to obesity, genes can also influence the development of postural control. It has been shown that some genes contribute to the performance of postural balance [39] and motor skills [40]. In our research, this variable was not included in the analysis; however, it is suggested that it be considered for future studies.

Within the limitations of this work, the indirect measurement of the PAL from a self-report questionnaire could be pointed out. Although the data related to the PAL were collected using validated instruments, they may not present the natural conditions of the participants, being susceptible to biases related to an overestimation of behaviors. Furthermore, the measurement of students exclusively from public schools could be a limitation, as it would have been interesting to know children’s behavior in the private education system.

## 5. Conclusions

In conclusion, in this study, it was possible to determine that overweight and obese children present a static postural balance deficit enhanced by variables such as gender, age, and anthropometric characteristics related to adiposity. In addition, in the dynamic balance, the PAL turns out to be a fundamental variable in the poor performance of children with overweight/obese nutritional status. Therefore, it is suggested that excess-weight children not only focus efforts on reducing body weight but also prioritize participating in physical activities to improve motor skills impaired by the effects of obesity.

## Figures and Tables

**Table 1 behavsci-13-00073-t001:** Basal characteristics of the sample (mean and standard deviations).

	Girls (n = 171)	Boys (n = 216)	Total (n = 387)
Age (years)	7.65 (1.12)	7.90 (1.12)	7.78 (1.12)
PAL	3.24 (0.91)	3.55 (0.96)	3.38 (0.95)
Body mass (kg)	33.89 (10.30)	34.32 (9.71)	34.13 (9.90)
Height (cm)	129.90 (9.73)	130.65 (9.41)	130.27 (9.48)
BMI (kg/m^2^)	19.82 (4.07)	18.81 (3.46)	19.81(3.71)
WC (cm)	63.05 (8.87)	64.62 (8.38)	63.92 (8.57)
WHR	0.83 (0.04)	0.88 (0.03)	0.85 (0.04)
Sum of skinfolds (mm)	101.05 (32.51)	97.77 (37.94)	99.51 (35.13)
Fat mass (%)	36.59 (4.92)	35.68 (5.64)	36.04 (5.31)
Muscle mass (%)	34.25 (4.26)	34.06 (4.11)	34.14 (4.15)
Residual mass (%)	11.05 (1.02)	11.72 (1.05)	11.42 (1.09)
Bone mass (%)	11.57 (0.97)	11.59 (1.36)	11.61 (1.21)
Skin mass (%)	6.88 (1.09)	6.85 (1.03)	6.86 (1.05)
Endomorph	5.65 (1.59)	5.26 (1.66)	5.49 (1.64)
Mesomorph	4.97 (1.38)	5.22 (1.19)	5.08 (1.28)
Ectomorph	1.59 (1.33)	1.46 (1.20)	1.52 (1.26)

PAL: physical activity level; BMI: body mass index; WC: waist circumference; WHR: waist–hip ratio.

**Table 2 behavsci-13-00073-t002:** Results of the static (center of pressure variables) and dynamic (performance in mSEBT) postural balance tests (mean and standard deviations).

	Girls (n = 171)	Boys (n = 216)	Total (n = 387)
*COP Variables*			
Velocity ML OE (m/s)	0.41 (0.20)	0.58 (0.52)	0.50 (0.42)
Velocity AP OE (m/s)	0.46 (0.14)	0.59 (0.26)	0.54 (0.22)
Mean velocity OE (m/s)	0.22 (0.01)	0.24 (0.02)	0.24 (0.20)
Area OE (m^2^)	0.02 (0.01)	0.02 (0.01)	0.02 (0.01)
Velocity ML CE (m/s)	0.50 (0.39)	0.50 (0.24)	0.50 (0.31)
Velocity AP CE (m/s)	0.71 (0.44)	0.77 (0.64)	0.75 (0.56)
Mean velocity CE (m/s)	0.27 (0.03)	0.28 (0.01)	0.27 (0.02)
Area CE (m^2^)	0.02 (0.01)	0.02 (0.01)	0.02 (0.01)
*mSEBT*			
Anterior (%)	58.55 (8.64)	60.12 (8.05)	59.82 (8.31)
PM (%)	76.12 (9.41)	82.01 (11.41)	79.02 (10.74)
PL (%)	60.18 (14.88)	66.23 (11.6)	63.57 (13.46)

ML: mediolateral; AP: anteroposterior; OE: open eyes; CE: closed eyes; PM: posteromedial; PL: posterolateral.

**Table 3 behavsci-13-00073-t003:** Significant multiple linear regression models obtained for the COP variables in the eyes-open test.

Variables	R^2^	Coefficient B	*p*	CI95%	
Velocity ML(m/s)	0.422				
Age		−0.231	0.001	−0.345	−0.117
Gender ^a^		0.227	0.037	0.015	0.440
Nutritional status ^b^		0.192	0.047	0.006	0.391
Sum of skinfolds		0.014	0.013	0.003	0.026
Fat mass		0.063	0.010	0.015	0.102
Velocity AP (m/s)	0.211				
Age		−0.074	0.039	−0.144	−0.004
Gender^a^		0.133	0.024	0.018	0.248
Mean velocity (m/s)	0.274				
Age		−0.005	0.022	−0.009	−0.001
Gender ^a^		0.013	0.006	0.004	0.022
Nutritional status ^b^		0.014	0.003	0.005	0.024

PAL: physical activity level; CI95%: confidence interval 95%; ML: mediolateral; AP: anteroposterior. ^a^ Gender: girls = 0, boys = 1. ^b^ Nutritional status: normal weight = 0, overweightness/obesity = 1.

**Table 4 behavsci-13-00073-t004:** Significant multiple linear regression models obtained for COP variables with eyes closed.

Variables	R^2^	Coefficient B	*p*	CI95%	
Velocity ML (m/s)	0.243				
Age		−0.075	0.026	−0.141	−0.009
Nutritional status ^a^		0.238	0.003	0.082	0.394
Velocity AP (m/s)	0.269				
Nutritional status ^a^		1.000	0.003	0.355	1.645
Ectomorph		−0.605	0.010	−1.064	−0.147
Mean velocity (m/s)	0.461				
Nutritional status ^a^		0.049	0.001	0.023	0.075
Fat mass		0.008	0.013	0.002	0.015
Endomorph		0.010	0.033	0.001	0.020
Ectomorph		−0.012	0.040	−0.025	−0.002

PAL: physical activity level; CI95%: confidence interval 95%; ML: mediolateral; AP: anteroposterior. ^a^ Nutritional status: normal weight = 0, overweightness/obesity = 1.

**Table 5 behavsci-13-00073-t005:** Significant multiple linear regression models obtained for dynamic postural balance.

Variables	R^2^	Coefficient B	*p*	CI95%	
Anterior (%)	0.389				
PAL		5.161	0.009	1.332	8.990
Nutritional status ^a^		−2.713	0.033	−5.195	−0.232
PM (%)	0.565				
PAL		6.544	0.013	1.458	11.630
WHR		−13.947	0.010	−19.549	−8.346
PL (%)	0.564				
PAL		12.540	0.001	6.850	18.231
Nutritional status ^a^		−6.545	0.031	−12.460	−0.630

PAL: physical activity level; CI95%: confidence interval 95%;. PM: posteromedial; PL: posterolateral. ^a^ Nutritional status: normal weight = 0, overweightness/obesity = 1.

## Data Availability

The data presented in this study are available upon request from the corresponding author. The data are not publicly available due to privacy.

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
