# Peer review of "Anthropometric Profile and Physical Activity Level as Predictors of Postural Balance in Overweight and Obese Children"

_behavsci, 2023, doi:10.3390/bs13010073_

Round 1

Reviewer 1 Report

The manuscript is an interesting read but raises important questions that require answering

1.     In the introduction, the authors state stats for obesity- Can the authors elaborate on the year- is this current, and where were the stats collected from?

2.     In the method section, the authors state that in the study data was collected from 387 school children (216 boys and 171 90 girls) between 6- and 9 years old- what was the ethnicity of the school children enrolled in the study. Were all the children from the same school?

3.     What is the BMI of the overweight students enrolled in the study?

4.     Are there previous studies that have looked at the correlation analysis between postural balance and obesity? How does this study compare with previous such studies?

5.     Since the study was conducted in pre-pubescent children, can the authors speculate on why they think there is a gender difference in the correlation analysis for postural imbalance for overweight and obese boys and girls?

6.     Do parents of kids that are obese and/or overweight suffer from postural imbalance and can data be collected to investigate the role of genetics in causing postural imbalance or is there pre-existing data for this?

7.     Can the authors comment more on the correlation between postural imbalance and obesity for children that were physically active? Is the postural imbalance primarily driven by physical inactivity or higher BMI? The authors need to tease these two factors apart.

Author Response

Dear reviewer, below we answer your questions and suggestions:

1. In the introduction, the authors state stats for obesity- Can the authors elaborate on the year- is this current, and where were the stats collected from? 

The information is updated and supported by recent research.

"Childhood overweight and obesity have increased over the last few decades: globally in 2016, an estimated 124 million children and adolescents aged 5–19 were affected by obesity, and 213 million were affected by overweight [1]. In Europe, overweight and obesity in schoolchildren are close to 20%, increasing in males [2]. In Latin America, the situation remains substandard and worse, showing close to 50% of overweight and obesity in children between 5-9 years of age [3]. This increase in childhood obesity is worrying since, due to projections, approximately 55% of obese children go on to be obese in adolescence, around 80% of obese adolescents will still be obese in adulthood and around 70% will be obese over age 30 [4]."   1. NCD Risk Factor Collaboration (NCD-RisC). Worldwide trends in body-mass index, underweight, overweight, and obesity from 1975 to 2016: a pooled analysis of 2416 population-based measurement studies in 128·9 million children, adolescents, and adults. Lancet 2017, 390, 2627-2642, doi:10.1016/S0140-6736(17)32129-3 2. Garrido-Miguel, M.; Oliveira, A.; Cavero-Redondo, I.; Álvarez-Bueno, C.; Pozuelo-Carrascosa, D.; Soriano-Cano, A.; Mar-tínez-Vizcaíno, V. Prevalence of Overweight and Obesity among European Preschool Children: A Systematic Review and Me-ta-Regression by Food Group Consumption. Nutrients 2019, 11, 1698, doi:10.3390/nu11071698. 3. Torres-González, EJ.; Zamarripa-Jáuregui, RG.; Carrillo-Martínez, JM.; Guerrero-Romero, F.; Martínez-Aguilar G. Preva-lence of overweight and obesity in school-age children. Prevalencia de sobrepeso y obesidad en niños escolares. Gac Med Mex 2020, 156, 182-186, doi:10.24875/GMM.M20000390 4. Simmonds, M.; Llewellyn, A.; Owen, CG.; Woolacott N. Predicting adult obesity from childhood obesity: a systematic review and meta-analysis. Obes Rev 2016, 17, 95-107, doi:10.1111/obr.12334   2.  In the method section, the authors state that in the study data was collected from 387 school children (216 boys and 171 90 girls) between 6- and 9 years old- what was the ethnicity of the school children enrolled in the study. Were all the children from the same school?   The ethnicity of the participants (Caucasian) and the number of schools from which the children were selected are added (Participants from 8 public schools were selected under a probabilistic stratified proportional sampling by sex and primary education courses (first, second, third, and fourth grade)).
.   3. What is the BMI of the overweight students enrolled in the study?   This information will be reported at the beginning of the results in the new version. "The BMI of the normal-weight schoolchildren was 15.86±0.74, of the overweight 18.47±0.87, and that of the obese 22.04±1.74"   4.  Are there previous studies that have looked at the correlation analysis between postural balance and obesity? How does this study compare with previous such studies?   In the discussion, some previous studies that have reported the relationship between postural balance and obesity are pointed out.   "Similar findings have previously been reported, demonstrating that nutritional status is correlated to static balance performance [21–23]. However, the interaction of other factors, such as body composition and somatotype, which could affect motor performance, was not considered. Dynamically, few studies have analyzed postural balance in overweight and obese children, reporting that excess weight could be correlated with poor balance performance [24,25]. In this research, it was possible to determine that the factors analyzed individually in previous reports, when considered together, can predict postural balance."   5. Since the study was conducted in pre-pubescent children, can the authors speculate on why they think there is a gender difference in the correlation analysis for postural imbalance for overweight and obese boys and girls?   In the discussion, an attempt is made to explain in a paragraph the difference found in the postural balance between boys and girls.

"In this research, gender was a determining factor in the models obtained for static postural balance, with males presenting a worse performance. This, together with an overweight/obese nutritional status, enhances poor balance test performance. Similar results were previously reported where it was determined that nutritional status and gender influence the deterioration of postural balance in children, with overweight and obese men being the most affected [23]. In the same way, age also became a factor that influences static postural balance, where younger children presented poor performance. It has been shown that women have a better balance at ages younger than 11-12 years than men [36]. It is believed that the greater motor hyperactivity observed in men of this age group would be the cause of a delay in the maturation of the vestibular system, which would directly affect the development of postural balance [37]. This would explain why gender and age determine static postural balance factors in overweight and obese children."

6.     Do parents of kids that are obese and/or overweight suffer from postural imbalance and can data be collected to investigate the role of genetics in causing postural imbalance or is there pre-existing data for this?   Unfortunately, the data that you indicate is not collected by us. We reviewed the literature and they have not been used in previous studies either. The idea you suggest is a very good one and we will certainly try to consider it in future studies.   7.  Can the authors comment more on the correlation between postural imbalance and obesity for children that were physically active? Is the postural imbalance primarily driven by physical inactivity or higher BMI? The authors need to tease these two factors apart.   The requested is added   "Our linear regression model showed that PAL was more influential than nutritional status in the performance of dynamic postural balance, which is related to what was previously reported where it is indicated that a low PAL can be riskier than excess weight in the physical function and health of people [36]"      

Reviewer 2 Report

The article is devoted to the study of predictors of postural balance in overweight and obese children.

I have a few suggestions for article minor revisions.

I suggest making the first introductory sentence in the abstract and then describing the purpose of the study. The authors give interesting statistics on the prevalence of obesity, but the sources are somewhat old. Is it possible to find newer articles to cite? Materials and methods, results, and discussion are described clearly and understandably. The conclusions could be expanded a little bit and made a little more detailed.

Author Response

The epidemiological data presented in the introduction were updated or supported by more recent studies.

The abstract and conclusion suggestions were made according to what you suggested.

Round 2

Reviewer 1 Report

The authors need to state the stats in the current year and not those for 2016. The data collected in 2016 is about a decade old.

The authors need to discuss the role of genes in maintaining postural balance in the discussion at-least.

Author Response

Epidemiological data are modified by those obtained from the world atlas of obesity 2022.

In the discussion, a paragraph is added that refers to the genetic factors that can influence postural balance.

thank you very much for your observations